# Quality of Life of Emirati Women with Cervical Cancer Using EORTC QLQ-30 and CX24: A First Look in the UAE

**DOI:** 10.3390/ijerph22050671

**Published:** 2025-04-24

**Authors:** Linda Smail

**Affiliations:** College of Interdisciplinary Studies, Zayed University, Dubai 19282, United Arab Emirates; linda.smail@zu.ac.ae

**Keywords:** cervical cancer, Emirati women, quality of life, health care policy, EORTC QLQ-CX24, EORTC QLQ-30

## Abstract

Background: Cervical cancer is the fourth leading cause of cancer-related mortality among women globally and remains a prevalent malignancy among Emirati women. This study assessed the quality of life of Emirati women with cervical cancer and identified key factors influencing their well-being to inform targeted interventions. Methods: A cross-sectional study was conducted among 72 Emirati women diagnosed with cervical cancer utilizing the Arabic-translated European Organization for Research and Treatment of Cancer Quality of Life Questionnaire (EORTC QLQ-C30 and QLQ-CX24). Sociodemographic and clinical data were collected. Statistical analyses included ANOVA, independent-sample *t*-tests, and, where assumptions were violated, Kruskal–Wallis and Mann–Whitney tests. Results: The mean global health status/QoL score was 64.4 (SD ± 20.4), indicating moderate well-being. The cognitive (69.9 ± 23.5) and role functioning (65.1 ± 25.0) scores were relatively high, whereas the social functioning score was lower (61.8 ± 25.2). Fatigue (41.5 ± 27.5), sleep disturbance (40.7 ± 31.3), and pain (39.4 ± 27.6) were the most prevalent symptoms. Radiotherapy negatively impacted sexual enjoyment (*p* = 0.019), whereas lower income and metastases were associated with worse symptom burden. Higher education, employment, and physical activity correlated positively with functional well-being. Conclusions: Early-stage diagnosis, financial stability, and physical activity were key predictors of better QoL. Addressing financial disparities, managing symptoms, and improving survivorship care are essential.

## 1. Introduction

Cervical cancer (CC) remains a major global health concern, ranking as the fourth most common cancer among women worldwide. According to the latest World Health Organization (WHO) data, approximately 660,000 new cases were diagnosed globally in 2022, resulting in around 350,000 deaths [1]. The burden of cervical cancer disproportionately affects women in low- and middle-income regions due to limited access to preventive measures such as HPV vaccination and regular screenings [2,3].

In the United Arab Emirates (UAE), cervical cancer is an important health issue for women despite its relatively low incidence compared with global figures. According to national cancer registry data, cervical cancer was the fifth most common cancer among Emirati women in 2021, with 141 newly reported cases (4.6% of all female cancers), and it is the sixth leading cause of cancer-related mortality [4,5]. The UAE has introduced preventive measures, including HPV vaccination programs for females aged 15–26 years and Pap smear screenings for women aged 25–65 years. These initiatives contributed to a notable reduction in in situ cases of cervical carcinoma between 2015 and 2017 [5]. Nevertheless, there remains a gap in comprehensive research and public awareness regarding cervical cancer prevention and management in the UAE. Elbarazi et al. highlighted the need for targeted educational interventions to improve both awareness and practices among Emirati women [6].

### 1.1. Incidence and Prevention of Cervical Cancer

Early detection and preventive measures are crucial in reducing the burden of CC worldwide [7]. In Asia, cure rates exceed 50%, with a one-year survival rate of 76.62% and a ten-year survival rate of 61.60% [3]. Among Asian countries, Kuwait has the highest CC survival rate, whereas Taiwan has the lowest. CC survivors commonly experience physical challenges such as abdominal discomfort, urine leakage, and menopause, along with mental health struggles related to body image and sexual well-being [8]. Variations in QoL among survivors have been observed across populations. Zhao et al. [9] reported that sociocultural factors significantly influence QoL disparities among Han individuals and ethnic minorities in Southwest China. Similarly, Prasongvej et al. [10] reported that CC survivors in Thailand experience notable QoL challenges compared with healthy women.

Cancer has a profound effect on various aspects of a patient’s life, including social, psychological, and sexual well-being. Studies indicate that these effects persist beyond treatment, with many CC survivors reporting sexual fears and disinterest in intimacy. Depression is also common and is often exacerbated by sexual dysfunction, such as dyspareunia and vaginal bleeding during intercourse [11]. However, social support has been shown to positively impact psychological well-being and overall patient conditions [12].

An analysis of CC cases among Gulf Cooperation Council (GCC) nationals from 1998 to 2012 revealed 2332 cases of invasive CC [13]. While the number of cases increased over time, the age-standardized incidence rate declined, indicating progress in prevention efforts. The peak age at diagnosis has also shifted toward older women. HPV is responsible for approximately 90% of CC cases, prompting some countries to introduce HPV vaccination programs targeting girls aged 10–20 years [14]. Educational initiatives related to HPV vaccination have been found to increase uptake rates and enhance CC prevention rates [15].

This study aims to explore the factors influencing the QoL of CC survivors and identify key areas for intervention. To date, no study has specifically examined the QoL of Emirati women with CC. This research is the first to investigate how sociodemographic and cultural factors impact QoL among Emirati CC survivors. The findings provide valuable insights that could inform healthcare policies and survivorship programs tailored to the needs of women in the UAE.

### 1.2. Impact of Cervical Cancer on Quality of Life

Quality of life (QoL) is a key concern for CC survivors, as both treatment and disease progression can significantly affect physical, psychological, and social well-being [16]. Psychological and behavioral factors play a major role in shaping the experiences of cancer patients, particularly women diagnosed with CC. Research has indicated that these women frequently report anger, frustration, and depression, which are often linked to sexual difficulties and infertility complications [11]. Educating young women about HPV vaccination has been shown to be crucial in preventing CC [15]. Additionally, CC screening before pregnancy is recommended, yet uptake remains low. A study in Japan revealed that only 18.8% of women had undergone CC screening within a year before pregnancy. Similarly, screening rates are low in Thailand, Turkey, the UAE, and India, where cultural and religious factors may limit opportunities for pre-pregnancy screening [17].

CC survivors often struggle with psychological and sexual well-being. The impact on sexual health varies depending on treatment type, with many survivors experiencing reduced sexual desire, infertility, vaginal atrophy, and cystitis. Combination treatments, such as surgery with radiation, tend to intensify these issues, leading to greater sexual dysfunction and dissatisfaction [18].

Spirituality also plays a significant role in QoL. Research has found a strong correlation between spiritual beliefs and emotional well-being in cancer patients. Patients with strong spiritual beliefs tend to experience greater emotional stability and reduced depression. Spirituality can also aid in managing pain and anxiety, leading to better physical health outcomes for cancer survivors [19].

Social support is another critical factor affecting QoL. Support from caregivers and the broader community can alleviate anxiety, fear, and the negative effects of treatment. Caregivers play a key role in providing relief and guidance throughout diagnosis and treatment [11,12]. Given its importance, cancer care should focus not only on treating tumors but also on enhancing QoL through psychological and social support [20].

### 1.3. Gaps in the Research and the Need for the Study

Research on the QoL of women with CC in Arab countries remains scarce. However, insights can be drawn from a Moroccan study that assessed the general and sexual QoL of 110 CC survivors [21]. While these survivors reported overall QoL comparable to that of healthy women, they experienced significant emotional challenges. They also reported lower satisfaction with sexual functioning and body image. The study identified social support and spiritual well-being as key predictors of QoL, accounting for 81% of the variance in QoL scores. The authors emphasized the need for healthcare professionals to address the complex relationships among QoL, CC treatment, and emotional well-being.

While region-specific studies remain limited, research from broader Asian contexts highlights several factors influencing QoL in women with CC [22]. Spagnoletti et al. [22] emphasized the importance of structured support systems for survivors, identifying the role of social, emotional, and medical support in shaping their QoL. Physical health conditions and their impact on daily activities have also emerged as key determinants of survivors’ well-being. Additionally, psychological distress, coping mechanisms, and cultural attitudes toward sexuality play significant roles in shaping survivors’ experiences.

Given the lack of studies on CC survivors in the UAE, more research is needed to examine how cultural, social, and economic factors influence their QoL. A deeper understanding of these factors would help develop effective interventions tailored to the needs of women in the region.

## 2. Methodology

### 2.1. Study Design and Sample

A community-based survey was conducted from April to October 2024 among Emirati women diagnosed with CC. Participants were recruited from Al Tawam, a major cancer referral center in Al Ain city operating under the Abu Dhabi Department of Health.

Eligible women (aged ≥ 18 years, Emirati nationality, and diagnosed specifically with CC) were contacted by phone before their scheduled hospital appointments by an Arabic-speaking nurse, who explained the study and gauged interest. Those who agreed were approached in person during their hospital visit to obtain written informed consent. The questionnaires were administered privately to ensure confidentiality, and no financial incentives were provided. Women diagnosed with other types of cancer were excluded. Given the study’s exploratory nature, recruitment was conducted throughout the study period without a predetermined sample size to ensure comprehensive data collection.

### 2.2. Ethics Approval and Consent to Participate

Ethical approval for this study was obtained from the Zayed University Research and Ethics Committee [ZU23_042_F] and the Al Tawam Human Research Ethics Committee [MF2058-2023-1011-DOH/ZHCD/2023/1322].

### 2.3. Data Collection and Study Instrument

The data were collected via a structured questionnaire that included three main parts:The first part, adapted from [23] with alterations made to certain items to better align with this study, contained items on both sociodemographic variables and reproductive characteristics. The sociodemographic data included the date of birth, level of education, marital status, employment status, and smoking habits. This part also collected data on QoL, such as the disease stage and the time elapsed since diagnosis.The second part included the European Organization for Research and Treatment of Cancer Quality of Life Core Questionnaire (EORTC QLQ-C30), a 30-item questionnaire designed to evaluate the physical, psychological, and social well-being of cancer patients. This questionnaire comprises nine scales: five functional scales, a global QoL scale, and three symptom scales (fatigue, pain, and nausea/vomiting). It also includes five single-item symptom scales (shortness of breath, sleep disturbance, loss of appetite, constipation, and diarrhea), as well as a final item that assesses the perceived financial impact of the disease. The first 28 items are rated on a scale from 1 (not at all) to 4 (very much), reflecting the present moment. For Items 29 (overall general health) and 30 (overall QoL), the responses range from 1 (very poor) to 7 (excellent), reflecting the past week.The third part included the CC-specific module (EORTC QLQ-CX24), a validated supplementary questionnaire module designed for CC patients. This module is intended to be used alongside the QLQ-C30 and includes three multi-item scales that assess symptom experience, body image, and sexual/vaginal functioning. Additionally, this module features six single items that assess lymphedema, peripheral neuropathy, menopausal symptoms, sexual worry, sexual activity, and sexual enjoyment.

The Arabic versions of the QLQ-C30 and the QLQ-CX24 were developed by the EORTC.

A professional translator translated the first part of the questionnaire from English to Arabic. A second bilingual speaker then compared the Arabic version with the English version word by word and translated it back into English. A panel of experts evaluated the Arabic version of the questionnaire to assess the readability, simplicity of the language, suitability of the items, and relationship of each item to the entire scale. The panel consisted of a professor specializing in obstetrics and gynecology, an oncologist, and a public health specialist. Their comments led to some changes being made. The internal consistency reliability of the Arabic version of the questionnaire was assessed using Cronbach’s coefficients. The coefficients were 0.907 and 0.867 for the EORTC QLQ-C30 and QLQ-CX24, respectively, indicating relatively good internal consistency.

### 2.4. Statistical Analysis

The data were coded, entered, and analyzed using the statistical package SPSS version 26 (SPSS, Chicago, IL, USA). Categorical data are presented as frequencies and percentages, whereas continuous data are presented as the means ± standard deviations (SDs).

The independent variables included sociodemographic information (age, education level, marital status, employment status, family income, smoking habits, and physical activity level).

We used the scoring guidelines for analyzing the EORTC scales [24,25,26].

The EORTC QLQ-C30 and EORTC QLQ-CX24 questionnaires were scored according to the official EORTC scoring manual [24,25,26]. Raw scores were calculated by averaging individual item responses with each scale, after which a linear transformation was applied to standardize these scores onto a scale ranging from 0 to 100.

The participants were categorized on the basis of their standardized scores as follows:-Functional scales/global QoL:○Scores ≥ 66.7 indicated good functioning/high QoL.○Scores < 33.3 indicated poor functioning/low QoL.-Symptom scales/items:○Scores ≥ 66.7 indicated severe/intense symptoms.○Scores < 33.3 indicated mild or minimal symptoms.

One-way analysis of variance (ANOVA) or the independent-sample *t*-test were carried out to test the equality of the population means across the categories of each independent variable (predictor) depending on the number of categories for the independent variables. If the statistical assumptions required for the one-way ANOVA and t tests were violated, nonparametric tests, namely, the Kruskal–Wallis test and Mann–Whitney test, were used instead. Additional exploration of the differences among the means was performed via post hoc analysis.

Pearson’s linear correlation coefficient was computed to assess the linear relationship between each of the outcome variables and each of the quantitative independent variables. The global health status/QoL scale score and the functional and symptom scale scores served as the dependent variables. The independent variables (age, time since diagnosis, marital status, educational level, employment status, income, parental status, radiotherapy status, chemotherapy status, physical activity level, and smoking habits) served as predictors for the models. Statistical tests with *p* values < 0.05 were considered statistically significant.

## 3. Results

### 3.1. Characteristics of the Study Sample

The final study sample included 72 Emirati women diagnosed with CC, with a mean age of 43.1 years (range 24–72). Most participants were diagnosed within the last two years, with an average time since diagnosis of 1.9 years. Two participants were excluded because of additional cancer diagnoses.

Educational levels varied, with most women having completed secondary education or higher (79.2%). The majority were married (76.4%), employed (56.9%), and had a monthly income below AED 10,000 (61.1%). Nearly half reported regular exercise (48.6%), and approximately 46% used contraceptives, with condoms being the most common method (39.4%).

Clinically, most participants were diagnosed at early stages (stage I: 30.6%; stage II: 20.8%). Chemotherapy was the predominant treatment method (40.3%), followed by combinations involving radiotherapy or surgery. Total hysterectomy was the most common surgical intervention (19.4%), with metastasis reported by approximately one-third (30.6%) of participants. Additional demographic details are provided in Table 1.

### 3.2. Quality of Life Scale Scores

The participants reported an overall moderate global health status/QoL (mean score: 64.4 ± 20.4), with approximately 36% indicating good global health (score ≥ 66.7).

In the functional domains of the QLQ-C30, cognitive functioning was notably the area with the highest score (69.9 ± 23.5), suggesting relatively few cognitive issues. Social functioning was rated lowest (61.8 ± 25.2), indicating particular social interaction challenges. The physical (64.3 ± 21.9), role (65.1 ± 25.0), and emotional functioning (65.2 ± 23.8) scores suggested moderate levels of day-to-day functioning.

Among the symptom scales, fatigue (41.5 ± 27.5) and sleep disturbances (insomnia, 40.7 ± 31.3) were among the highest-rated issues, reflecting their notable impact on participants’ daily lives. Pain was also significant, with a mean score of 39.4 ± 27.6. Less prevalent but noteworthy symptoms included financial difficulties (mean score: 38.4 ± 34.8) and constipation (mean score: 36.1 ± 34.8).

On the cervical cancer-specific scale (QLQ-CX24), symptom experience scores indicated moderate symptom burden (29.6 ± 17.3). Body image concerns (39.2 ± 31.3) and sexual/vaginal functioning (29.5 ± 27.8) revealed significant psychosocial and sexual health impacts, respectively. Notably, sexual activity (mean: 26.9 ± 27.2) and sexual enjoyment (mean: 27.7 ± 30.5) scores were relatively low, highlighting specific areas of sexual health needing attention. These results collectively indicate that while many women maintain moderate QoL, substantial subsets face challenges, particularly those related to fatigue, pain, sleep issues, social interaction, and sexual functioning. The comprehensive scores and additional details are summarized in Table 2.

### 3.3. Factors Associated with Quality of Life Scale Scores

Several participant characteristics were significantly associated with variations in QoL scores. Higher monthly income (>AED 30,000) was strongly linked to better global health/QoL (*p* = 0.002). Similarly, participants who underwent radiotherapy reported significantly higher global health/QoL scores (*p* = 0.0115) (Table 3).

The physical functioning scores were notably better in women without metastasis (*p* < 0.001) and in those treated with radiotherapy (*p* = 0.008). Similarly, role functioning improved significantly with radiotherapy (*p* = 0.021) and the absence of metastasis (*p* = 0.016). Regular physical activity significantly enhanced emotional (*p* = 0.007) and social functioning (*p* = 0.026). Cognitive functioning was greater among employed women (*p* = 0.028) and those without metastasis (*p* = 0.029). Social functioning was notably better in employed women (*p* = 0.046) and participants with early-stage disease (*p* = 0.047).

In terms of symptoms, fatigue and pain were significantly lower among physically active women (fatigue: *p* = 0.018; pain: *p* = 0.005). Participants with higher incomes reported significantly less nausea (*p* = 0.028), appetite loss (*p* = 0.015), diarrhea (*p* = 0.008), and financial difficulties (*p* = 0.001). Additionally, dyspnea was notably more common in advanced-stage patients (*p* = 0.021), and less common in those receiving radiotherapy (*p* = 0.004). These results clearly emphasize the influence of socioeconomic status, treatment type, disease severity, and physical activity level on various dimensions of QoL in Emirati women with cervical cancer. Further detailed statistics are available in Table 3 and Table 4.

### 3.4. Symptom Items on the QLQ-CX24

Analysis of symptom experiences and body image (QLQ-CX24) revealed key associations with participant characteristics (Table 5). Notably, physically inactive (*p* < 0.001) and unemployed participants (*p* = 0.046) reported significantly greater symptom burdens. A higher monthly income (>AED 30,000) was associated with fewer reported symptoms (*p* = 0.011). Additionally, participants with metastasis experienced a significantly greater symptom burden (*p* = 0.044).

Age significantly influenced sexual/vaginal functioning (*p* = 0.005) and sexual worry (*p* = 0.003), with younger participants (35–44 years) reporting better sexual functioning but greater sexual worry. Marital status influenced sexual worry (*p* = 0.046), as divorced participants reported less worry than single or married individuals did. Employment (*p* = 0.042) and chemotherapy (*p* = 0.017) were associated with increased sexual worry.

Lymphedema was significantly worse among physically inactive participants (*p* = 0.011), lower-income participants (*p* = 0.015), and those with metastasis (*p* = 0.037). Similarly, the incidence of peripheral neuropathy was greater in physically inactive participants (*p* = 0.003) and older participants (*p* = 0.049).

These results highlight the critical role of physical activity, socioeconomic factors, and disease severity in managing symptom burdens among cervical cancer survivors. The detailed statistics can be found in Table 5.

### 3.5. Functional Items on the QLQ-CX 24

The results in Table 6 provide a detailed analysis of the functional items on the QLQ-CX24, including sexual activity and enjoyment, across various demographic, socioeconomic, and clinical characteristics. The findings showed that the sexual activity scores varied significantly across age groups (*p* = 0.023). Women aged 24–44 years had the highest levels of sexual activity, whereas those aged 54–72 years presented the lowest scores. Similarly, education level was a key determinant, with university graduates exhibiting significantly higher sexual activity scores than participants with lower education levels (*p* = 0.010). Employment status also showed a significant relationship, with employed women reporting better sexual activity than unemployed women did (*p* = 0.010). Additionally, smoking status was an influencing factor, with smokers demonstrating significantly higher sexual activity scores than nonsmokers and those who had quit smoking (*p* = 0.044). The participants who did not receive radiotherapy had lower sexual activity scores than did those who did receive radiotherapy (*p* = 0.029).

With respect to sexual enjoyment, no significant differences were observed across most variables. However, women who received radiotherapy presented significantly lower sexual enjoyment than did those who did not receive radiotherapy (*p* = 0.019). This decline in sexual enjoyment among radiotherapy recipients may be attributed to treatment-related side effects such as vaginal dryness, fibrosis, and reduced vaginal elasticity, which can lead to pain and discomfort during intercourse.

### 3.6. Predictors of Quality of Life

The results of the multiple regression analysis for predicting the global health status of CC survivors are presented in Table 7. The dependent variable is the global health status, while the independent variables included marital status, education level, employment status, income, cancer stage, time since diagnosis, and treatment modalities such as radiotherapy and chemotherapy.

Income was the most influential predictor of QoL, demonstrating a significant and strong positive association with global health status (β = 0.502, SE = 2.462, *p* < 0.001). Higher income levels were associated with increased QoL, with substantial disparities across income categories. Survivors with an income of less than AED 10,000 had significantly lower QoL scores (β = −34.112, SE = 7.2183, *p* < 0.001) than those earning more than AED 10,000.

The overall regression model explained 17% of the variance in global health status, as reflected by the adjusted R-square value (R^2^ = 0.170). The model was statistically significant (F = 2.815, *p* = 0.010), indicating that the predictors collectively positively influence QoL among CC survivors. However, the relatively low R^2^ value underscores the potential contributions of additional unmeasured factors, such as psychological health, social support, and lifestyle behaviors, in shaping QoL.

## 4. Discussion, Implications, and Recommendations

This study, which uses the EORTC QLQ-C30 and QLQ-CX24, provides the first detailed insight into factors influencing the QoL of Emirati women with CC, identifying key areas for support and intervention. The findings highlight the substantial impact of socioeconomic, demographic, and clinical factors on survivors’ well-being. These findings emphasize the need for targeted interventions to address the multifaceted burden of CC survivorship.

The moderate global health score (64.4 ± 20.4) aligns with international findings among CC survivors, although it highlights particular vulnerabilities within the Emirati context. Factors such as income, physical activity, and education level emerged as key influencers. Consistent with previous studies [9,10], higher income and education levels were positively associated with increased functional scores and reduced symptom burden. Stable employment also contributes to better QoL, further underscoring the role of socioeconomic stability in mitigating the challenges posed by CC [27].

Sociodemographic factors, including education, employment, and income levels, emerged as significant predictors of QoL. Women with higher educational attainment and stable employment had better functional and symptom scores, emphasizing the importance of socioeconomic stability [28]. However, the financial burden of CC treatment, particularly for lower-income women (β = 9.365, *p* < 0.001), underscores the need for targeted financial support to reduce economic stress and improve access to care, as global studies have emphasized [29,30].

Symptom-specific analysis revealed that fatigue, pain, and sleep disturbances were the most prominent issues, echoing trends in broader Asian populations [22]. These findings—consistent with global studies—suggest that routine screening for fatigue and sleep disturbances should be integrated into survivorship care plans, along with tailored interventions such as fatigue management programs, sleep therapy, and psychosocial counseling. Remarkably, radiotherapy and chemotherapy are linked to decreased QoL, emphasizing the need for targeted interventions to mitigate treatment-related side effects. Emotional and sexual functioning, as assessed by the QLQ-CX24, were particularly impacted, with survivors reporting significant distress related to body image, societal perceptions, and sexual dysfunction (low sexual activity and enjoyment scores). These findings reflect global challenges in addressing posttreatment sexual health and body image concerns among CC survivors [18,29].

This aligns with the literature identifying sexual health as a critical yet often overlooked dimension of survivorship care [31,32]. Emotional well-being was also compromised, with participants reporting significant distress related to body image and societal perceptions. Given the sociocultural sensitivities surrounding discussions of sexual health in the UAE, these findings highlight the need for discreet, culturally appropriate sexual health counseling services for CC survivors. Clinicians should be equipped to address these concerns within a culturally sensitive framework, ensuring that survivors receive the necessary support without stigma or discomfort.

Clinical factors such as the cancer stage and treatment modality significantly influence QoL outcomes [28]. Women diagnosed at earlier stages have better QoL scores, highlighting the importance of early detection and timely treatment [32]. Furthermore, although radiotherapy is commonly linked with reduced QoL, this study revealed that patients who received radiotherapy alongside early-stage diagnosis had better outcomes [31]. These findings reinforce the need for comprehensive early detection programs, as timely intervention not only improves survival rates but also enhances long-term QoL.

Unlike in Western studies, where marital and employment status strongly influence QoL, this study found no significant associations. This may be due to the protective role of extended family and community networks in the UAE, which buffer the impact of socioeconomic stressors. In many Arab societies, family members provide substantial financial, emotional, and caregiving support, potentially reducing the direct impact of employment on a patient’s ability to access care. Additionally, marital status may not play a significant role in QoL because of strong familial ties and community support systems that remain available regardless of a woman’s marital situation. Future research should explore these social determinants further to develop culturally appropriate survivorship programs. Additionally, healthcare policies in the UAE should leverage these existing social structures while also ensuring that support is available for women who may not have strong familial networks.

### 4.1. Practical Implications for Practitioners

Healthcare providers are critical in delivering holistic care to CC survivors. This means that the adoption of a patient-centered approach that integrates medical, psychological, and social support into routine care is important. The significant symptom burden suggests the need for tailored symptom management strategies, with clinicians prioritizing early intervention to increase QoL. Additionally, sexual health has emerged as a particularly challenging area, requiring specialized counseling and intervention services. Practitioners should also leverage culturally sensitive communication strategies to address these issues in a way that respects the values and beliefs of Emirati women.

### 4.2. Policy Implications

A critical implication of this study is the need for expanded HPV vaccination and screening programs. While the UAE has made strides in implementing these initiatives, greater outreach is necessary to increase coverage, particularly in underserved populations. Financial barriers to care have also been highlighted as a significant concern. Policymakers should consider implementing subsidized care models or expanding insurance coverage to alleviate the economic burden on CC survivors.

In addition, this study underscores the importance of integrating psychosocial support into cancer care policies. This involves mandating the inclusion of mental health professionals and social workers in oncology teams and funding community-based support programs. Addressing emotional well-being, body image concerns, and societal stigma requires a coordinated policy effort that bridges healthcare, education, and social services.

Future research should focus on evaluating the effectiveness of these policy interventions in improving QoL outcomes for CC survivors. Longitudinal studies tracking the impact of enhanced screening, financial support, and psychosocial care programs will be essential in refining public health strategies and ensuring sustainable improvements in survivorship care.

### 4.3. Implications for Stakeholders

Nongovernmental organizations, community leaders, and educational institutions are vital in increasing awareness and promoting early detection of CC. Community-based interventions, such as culturally tailored educational campaigns, can significantly influence health outcomes. These campaigns should focus on destigmatizing CC and encouraging preventive behaviors, including HPV vaccination and regular screenings. Furthermore, fostering partnerships between healthcare providers and community stakeholders can help disseminate information more effectively and build trust among target populations.

### 4.4. Healthcare Policy and System Interventions

Enhanced screening programs, which focus on expanding access to Pap smear tests and HPV vaccinations, are essential to improve early detection rates. Financial support mechanisms should be implemented through subsidies or insurance schemes, particularly those that target low-income patients, to ensure equitable access to care. The development of integrated care models and the incorporation of multidisciplinary teams that include oncologists, psychologists, and social workers to provide comprehensive patient support are crucial. These initiatives should be supported by robust policy frameworks that ensure sustainable implementation and monitoring of outcomes.

### 4.5. Community and Patient-Centered Strategies

Culturally tailored education programs must be developed to address prevention, treatment, and survivorship while considering the cultural and religious sensitivities of Emirati women. This includes establishing comprehensive peer support groups that provide emotional and practical assistance to survivors throughout their journey. Physical activity and nutrition programs should be specifically tailored to meet the needs of CC survivors, accounting for their physical capabilities and cultural preferences. These programs should be integrated into healthcare structures to ensure accessibility and sustained participation.

While January is recognized as the Cervical Cancer Awareness Month in the UAE, efforts to combat CC must extend beyond a single month. Year-round initiatives are essential to sustain awareness, promote preventive behaviors, and ensure continuous access to screening and vaccination programs. Awareness campaigns should be integrated into broader healthcare strategies, with a focus on consistent community engagement and policy support to achieve long-term reductions in CC incidence and mortality.

### 4.6. Addressing Income Disparities in the QoL of Cervical Cancer Survivors

The findings from this study underscore the significant impact of income disparities on QoL among Emirati CC survivors. Addressing these disparities is essential for improving health outcomes and mitigating inequities in cancer care. To achieve this, several policy recommendations have been proposed to create a more inclusive and supportive healthcare system.

The UAE government and healthcare systems should implement financial support mechanisms to provide subsidized or free healthcare services for low-income CC survivors. This support should include costs associated with routine follow-ups, diagnostic tests, treatments, and palliative care. With reduced financial healthcare burdens, survivors can access necessary services without jeopardizing their financial stability.

Integrating economic support into cancer care models is another critical step. Financial counseling and social support services should be embedded within cancer care plans to assist survivors in navigating income-related challenges. Hospitals and clinics, both public and private, can employ financial aid officers to help patients access available government or nonprofit assistance programs. This integration can empower survivors to seek financial help proactively and reduce the stress of navigating complex support systems.

Expanding existing health insurance to include comprehensive coverage for CC survivors is also essential. Health insurance should address not only treatment costs but also nonmedical expenses, such as expenses related to rehabilitation, mental health counseling, and employment reintegration programs, which significantly impact survivors’ QoL. In addition, employment assistance programs should be established to support survivors’ reintegration into the workforce. These programs can offer job training, skill development, and workplace accommodations, fostering economic independence.

Local governments and community organizations can play pivotal roles by developing community-based support initiatives. Establishing centers that provide free or low-cost services, such as legal aid, financial planning, and social programs, can improve survivors’ QoL and bridge the gap in service accessibility. Additionally, public awareness campaigns are essential to highlight the economic challenges faced by CC survivors. Advocacy for equitable healthcare access can garner public and institutional support, driving the implementation of policies aimed at reducing income-based disparities.

To ensure the success of these interventions, monitoring and evaluation systems should be established. To identify gaps in current policies, government and healthcare institutions must track the financial and health outcomes of CC survivors. Data-driven adjustments can enable the optimization of resources and enhance the impact of support programs. Future research should explore the long-term impact of financial assistance programs on the QoL of CC survivors, assessing whether targeted interventions, such as subsidized care or employment reintegration initiatives, lead to sustained improvements in health and well-being. Additionally, comparative studies between high-income and low-income CC survivors in the UAE could provide deeper insights into how economic disparities shape health outcomes and treatment adherence. Identifying the most effective financial and social support interventions will be crucial for informing future policy development.

This study has several limitations that should be acknowledged. First, as a cross-sectional study, it captures QoL at a single point in time, limiting the ability to assess long-term changes and the evolving impact of financial disparities on survivorship outcomes. Second, while this study focused on Emirati women, the findings may not be fully generalizable to expatriate populations within the UAE, who may face different economic and healthcare challenges. Finally, self-reported data on financial hardship and QoL may be subject to response biases. Despite these limitations, this study provides valuable insights into the economic determinants of QoL in CC survivors and highlights critical areas for policy intervention.

## 5. Conclusions and Future Directions

This pioneering study provides the first comprehensive examination of the QoL challenges faced by Emirati women with CC, highlighting the interplay among clinical, socioeconomic, and cultural factors in shaping survivor outcomes. The findings emphasize the critical roles of socioeconomic stability, physical and emotional health, and cultural considerations, underscoring the need for integrated care strategies tailored to this population. By identifying key areas for intervention—symptom management, sexual health, financial support, and culturally sensitive care—this research lays the groundwork for targeted initiatives that can significantly improve the QoL of CC survivors in the UAE.

Future research should prioritize longitudinal studies to track the long-term impacts of CC and its treatments, assess changes in QoL over time, and evaluate the effectiveness of interventions. Expanding sample sizes across diverse UAE populations, including expatriates and marginalized groups, will improve the generalizability of findings and inform equitable healthcare policies. Comparative studies across Arab countries could highlight regional disparities, while qualitative approaches would offer deeper insights into survivors’ lived experiences.

Further research is also needed to refine intervention strategies, particularly in psychosocial counseling, financial support programs, and culturally adapted educational campaigns. The validation of QoL assessment tools tailored to Arab women will ensure accurate and culturally relevant patient outcome measurements. Additionally, investigating the role of religious beliefs, social support networks, and cultural expectations in shaping survivorship experiences will provide a more holistic understanding of CC survivorship in the UAE.

By addressing these research priorities, future studies can refine strategies for improving survivorship care and reducing the burden of CC in the region. These efforts will contribute to a more equitable healthcare system, ensuring evidence-based practices and policies that enhance the well-being of CC survivors.

## Figures and Tables

**Table 1 ijerph-22-00671-t001:** Characteristics and therapeutic situations of the participants (N = 72).

Variable	Cervical Cancer Survivors, *n* (%)
Age (years), mean	43.1 ± 11.4
24–34	15 (20.8)
35–44	31 (43.1)
45–54	12 (16.7)
54–72	14 (19.4)
Education level	
Illiterate	3 (4.2)
Primary and preparatory school	12 (16.7)
Secondary school	18 (25)
University undergraduate	29 (40.3)
University postgraduate	10 (13.9)
Marital status	
Single	10 (13.9)
Married	55 (76.4)
Divorced	7 (9.7)
Employment status	
Employed	41 (56.9)
Not employed (1 retired)	31 (43.1)
Smoking habits	
Smoker	13 (18)
Nonsmoker	55 (76.4)
Previous smoker	4 (5.6)
Family income (AED)	
≤10,000 (USD 2500)	44 (61.1)
10,000–20,000 (USD 2500–5000)	10 (13.9)
20,000–30,000 (USD 5000–7500)	9 (12.5)
≥30,000 (>USD 7500)	9 (12.5)
Exercise status	
Yes	35 (48.6)
Contraception use	
Yes	33 (45.8)
No	39 (54.2)
Contraceptive used	
Hormonal contraceptive pills	10 (30.3)
Condoms	13 (39.4)
Intrauterine device	10 (30.3)
Family history of cancer/CC	
Yes	19 (26.4)
No	44 (61.1)
Don’t know	9 (12.5)
Time since diagnosis (years), mean	1.9 ± 2.9
<1	19 (26.4)
1–2	35 (48.6)
2–3	9 (12.5)
3–4	3 (4.2)
4–5	3 (4.2)
>5	3 (4.2)
Cancer stage	
0	9 (12.5)
I	22 (30.6)
II	15 (20.8)
III	13 (18.1)
IV	5 (6.9)
Don’t know	8 (11.1)
Age at diagnosis (years), mean	41.5 ± 11.1 (Range 23–72)
≤29	7 (9.7)
30–39	31 (43.1)
40–49	21 (29.2)
50–59	7 (9.7)
≥60	6 (8.3)
Treatment method	
Radiotherapy (RT)	8 (11.1)
Chemotherapy (including tablets)	30 (41.7)
Surgery	6 (8.3)
Hormonal therapy (HT)	2 (2.8)
Chemotherapy, HT	1 (1.4)
Chemotherapy, RT	9 (12.5)
Chemotherapy, RT, HT	1 (1.4)
Chemotherapy, RT, surgery	7 (9.7)
Does not know	8 (11.1)
Surgery type	
Total hysterectomy	14 (19.4)
Radical hysterectomy	4 (5.6)
Trachelectomy	3 (4.2)
Pelvic exenteration	7 (9.7)
Partial hysterectomy, excluding the cervix	5 (6.9)
Other ^a^	17 (23.6)
Metastasis to other parts of the body	
Yes	22 (30.6)
No	40 (55.6)
Don’t know	10 (13.9)

^a^ Other included combinations of treatments such as chemotherapy and HT; surgery combined with chemotherapy and targeted therapy; surgery combined with RT, HT, and targeted therapy; and surgery combined with chemotherapy, HT, and targeted therapy. Additionally, it could include experimental treatment protocols or, less commonly, surgical interventions tailored to individual patient cases.

**Table 2 ijerph-22-00671-t002:** Mean scores for all items on the QLQ-C30 and QLQ-CX24 (N = 72).

Variables	Items, *n*	Mean ± SD	95% CI	Score, *n* (%)	
				<33.3 ^a^	≥66.7 ^b^
QLQ-C30					
Global health status/QoL	2	64.4 ± 20.4	59.6–69.2	1 (1.4)	26 (36.1)
Functional scales ^b^					
Physical functioning	5	64.3 ± 21.9	59.1–69.4	6 (8.3)	32 (44.4)
Role functioning	2	65.1 ± 25.0	59.2–70.9	4 (5.6)	22 (30.6)
Emotional functioning	4	65.2 ± 23.8	59.6–70.8	6 (8.3)	29 (40.3)
Cognitive functioning	2	69.9 ± 23.5	64.4–75.4	2 (2.8)	32 (44.4)
Social functioning	2	61.8 ± 25.2	55.9–67.7	3 (4.2)	17 (23.6)
Symptom scales ^c^					
Fatigue	3	41.5 ± 27.5	35.0–48.0	26 (36.1)	9 (12.5)
Nausea and vomiting	2	26.6 ± 26.8	20.3–32.9	35 (48.6)	3 (4.2)
Pain	2	39.4 ± 27.6	32.9–45.8	19 (26.4)	9 (12.5)
Dyspnea	1	25.5 ± 28.8	18.7–32.2	34 (47.2)	3 (4.2)
Sleep disturbance—insomnia	1	40.7 ± 31.3	33.4–48.1	17 (23.6)	8 (11.1)
Appetite loss	1	30.6 ± 32.5	22.9–38.2	30 (41.7)	7 (9.7)
Constipation	1	36.1 ± 34.8	27.9–44.3	26 (36.1)	10 (13.9)
Diarrhea	1	22.2 ± 31.1	14.9–29.5	41 (56.9)	6 (8.3)
Financial impact	1	38.4 ± 34.8	30.3–46.6	22 (30.6)	12 (16.7)
QLQ-CX24					
Symptom scales ^c^					
Symptom experience	11	29.6 ± 17.3	25.5–33.7	34 (47.2)	1 (1.4)
Body image	3	39.2 ± 31.3	31.9–46.5	25 (34.7)	10 (13.9)
Sexual/vaginal functioning (*n* = 44)	4	29.5 ± 27.8	21.1–38.0	25 (34.7)	5 (6.9)
Lymphedema	1	25.5 ± 32.4	17.9–33.1	39 (54.2)	5 (6.9)
Peripheral neuropathy	1	29.6 ± 30.4	22.5–36.8	29 (40.3)	5 (6.9)
Menopausal symptoms	1	33.3 ± 28.0	26.8–39.9	20 (27.8)	5 (6.9)
Sexual worry	1	31.9 ± 32.8	24.2–39.7	29 (40.3)	7 (9.7)
Functional scales ^b^					
Sexual activity	1	26.9 ± 27.2	20.5–33.2	31 (43.1)	1 (1.4)
Sexual enjoyment (*n* = 47)	1	27.7 ± 30.5	18.7–36.6	22 (30.6)	2 (2.8)

^a^ For the functional scales, women with scores < 33.3% had problems; those with scores ≥ 66.7% had good functioning. For the symptom scales/symptoms, women with scores <33.3% had good functioning; those with scores ≥66.7% had problems. ^b^ For the functional scales, higher scores indicate better functioning. ^c^ For the symptom scales, higher scores indicate worse functioning.

**Table 3 ijerph-22-00671-t003:** Global health and functional scales of the QLQ-C30 by independent variables (N = 72) ^a^.

Characteristics	Global Health Status/Qol ^b^	Physical Functioning ^b^	Role Functioning ^b^	Emotional Functioning ^b^	Cognitive Functioning ^b^	Social Functioning ^b^
Age (years), mean ± SD						
24–34	63.3 ± 19.9	58.7 ± 16.8	64.4 ± 21.7	61.1 ± 21.5	70.0 ± 23.7	64.4 ± 28.8
35–44	65.1 ± 17.0	69.5 ± 18.4	67.7 ± 20.2	72.3 ± 19.6	74.7 ± 21.0	65.1 ± 21.2
45–54	68.1 ± 20.0	67.8 ± 18.2	62.5 ± 31.1	61.8 ± 28.7	58.3 ± 20.7	54.2 ± 26.7
54–72	60.7 ± 28.6	55.7 ± 32.6	61.9 ± 33.6	56.5 ± 28.0	69.0 ± 29.1	58.3 ± 29.1
*p*-value	0.869	0.319	0.922	0.231	0.192	0.574
Time since diagnosis (years), mean ± SD						
<1	60.5 ± 15.7	62.1 ± 21.3	60.5 ± 23	66.7 ± 21.7	71.1 ± 16.5	58.8 ± 26.3
1–2	60.2 ± 20.4	68 ± 18.8	69.5 ± 18.3	63.3 ± 24.9	69.0 ± 24.3	66.7 ± 21.8
2–3	68.5 ± 22.0	58.5 ± 31.1	66.7 ± 40.8	66.7 ± 28.6	77.8 ± 23.6	51.9 ± 33.8
3–4	80.6 ± 17.3	66.7 ± 29.1	55.6 ± 38.5	63.9 ± 33.7	50 ± 44.1	44.4 ± 19.2
4–5	52.8 ± 45.9	60.0 ± 17.6	50.0 ± 44.1	77.8 ± 17.3	72.2 ± 34.7	72.2 ± 34.7
>5	61.1 ± 19.2	53.3 ± 35.3	61.1 ± 25.5	61.1 ± 17.3	66.7 ± 28.9	61.1 ± 25.5
*p*-value	0.639	0.861	0.677	0.926	0.803	0.441
Marital status, mean ± SD						
Single	59.2 ± 17.3	62.0 ± 17.8	68.3 ± 18.3	53.3 ± 27	65.0 ± 28.8	51.7 ± 29.9
Married	65.5 ± 18.7	64.2 ± 22.9	65.8 ± 25.3	68.2 ± 21.9	72.1 ± 21.8	62.7 ± 24.8
Divorced	63.1 ± 35.6	67.6 ± 20.9	54.8 ± 31.5	58.3 ± 30.4	59.5 ± 28.6	69.0 ± 20.2
*p*-value	0.606	0.834	0.551	0.192	0.400	0.386
Education level, mean ± SD						
Illiterate	50.0 ± 16.7	57.8 ± 27.8	44.4 ± 41.9	55.6 ± 34.7	66.7 ± 16.7	55.6 ± 48.1
Primary and preparatory school	56.9 ± 26.8	62.2 ± 26.7	54.2 ± 30.3	56.3 ± 27.1	58.3 ± 27.1	52.8 ± 25.5
Secondary school	65.3 ± 20.1	63.3 ± 18.5	72.2 ± 21.4	66.7 ± 20.6	73.1 ± 23	63 ± 23.3
University undergraduate	67.2 ± 18.5	68.3 ± 17.4	70.1 ± 18.6	65.5 ± 21.6	73.6 ± 20.2	64.9 ± 21.5
University graduate	67.5 ± 18.6	58.7 ± 32.2	56.7 ± 30.6	75 ± 28.6	68.3 ± 29.9	63.3 ± 33.1
*p*-value	0.511	0.859	0.373	0.400	0.491	0.526
Employment, mean ± SD						
Yes	66.7 ± 18.3	65.7 ± 21.3	69.5 ± 22.9	67.3 ± 23.6	75.2 ± 21.4	66.7 ± 23
No	61.3 ± 22.9	62.4 ± 22.8	59.1 ± 26.8	62.4 ± 24.2	62.9 ± 24.6	55.4 ± 27
*p*-value	0.310	0.563	0.112	0.344	0.028	0.046
Monthly income (AED), mean ± SD						
<10,000	58.5 ± 19.3	63.9 ± 22.6	59.5 ± 26.8	61.0 ± 22.3	66.7 ± 23.8	58.3 ± 27.5
10,000–20,000	68.3 ± 17.9	62 ± 19.9	68.3 ± 21.4	72.5 ± 22.6	78.3 ± 15.8	66.7 ± 15.7
20,000–30,000	65.7 ± 19.7	56.3 ± 22.6	72.2 ± 20.4	66.7 ± 22.8	66.7 ± 23.6	57.4 ± 25.2
>30,000	87 ± 12.6	76.3 ± 17.7	81.5 ± 15.5	75.9 ± 31.3	79.6 ± 27.4	77.8 ± 16.7
*p*-value	0.002	0.271	0.079	0.089	0.197	0.192
Physical activity, mean ± SD						
Yes	68.3 ± 19.5	69.7 ± 18	71.4 ± 17.9	71.7 ± 22.1	77.1 ± 22.2	69 ± 19
No	60.6 ± 20.8	59.1 ± 24.1	59 ± 29.3	59 ± 24.1	63.1 ± 23	55 ± 28.6
*p*-value	0.162	0.101	0.061	0.007	0.007	0.026
Smoking habits, mean ± SD						
Smoker	55.8 ± 12.4	63.1 ± 17.3	59 ± 25.1	73.7 ± 16.6	66.7 ± 19.2	64.1 ± 20.2
Nonsmoker	65.2 ± 21.5	64.4 ± 23.7	66.7 ± 25.9	63.2 ± 25.6	70.9 ± 25.1	60.9 ± 26.5
Previous smoker	81.3 ± 14.2	66.7 ± 5.4	62.5 ± 8.3	64.6 ± 10.5	66.7 ± 13.6	66.7 ± 27.2
*p*-value	0.052	0.830	0.462	0.386	0.740	0.921
Children, mean ± SD						
Yes	65.8 ± 21.2	65.9 ± 22.4	65.5 ± 26.3	66.5 ± 24.3	70.4 ± 23.6	62.6 ± 24.6
No	58.3 ± 16.3	57.6 ± 18.8	63.1 ± 19.8	59.5 ± 21.6	67.9 ± 24	58.3 ± 28.3
*p*-value	0.166	0.095	0.529	0.292	0.652	0.605
Stage, mean ± SD						
0	57.4 ± 20.2	59.3 ± 22	53.7 ± 29.8	63.0 ± 30.9	61.1 ± 16.7	44.4 ± 33.3
I	66.7 ± 18	65.5 ± 20.7	70.5 ± 23	65.5 ± 18.1	68.9 ± 24.3	61.4 ± 19.5
II	57.8 ± 22.8	68.4 ± 17.5	68.9 ± 23.5	71.1 ± 19.1	80 ± 20.1	65.6 ± 21.3
III	69.9 ± 19.1	61.5 ± 20.9	57.7 ± 20	65.4 ± 26.1	61.5 ± 24.9	71.8 ± 21.9
IV	71.7 ± 13.9	54.7 ± 34.4	40 ± 25.3	41.7 ± 20.4	53.3 ± 21.7	40.0 ± 27.9
*p*-value	0.313	0.812	0.063	0.214	0.087	0.047
Radiotherapy, mean ± SD						
Yes	69 ± 27.3	70.7 ± 25.8	72 ± 31.1	67.7 ± 28.7	74.7 ± 24.1	67.3 ± 29.5
No	61.9 ± 15.4	60.9 ± 18.9	61.3 ± 20.6	63.8 ± 21	67.4 ± 23	58.9 ± 22.5
*p*-value	0.0115	0.008	0.021	0.381	0.192	0.118
Chemotherapy, mean ± SD						
Yes	65.8 ± 18.1	66.7 ± 22.2	66.3 ± 25.9	68.4 ± 23.2	74.5 ± 22.2	64.2 ± 24.6
No	61.7 ± 24.4	59.7 ± 20.9	62.7 ± 23.7	59 ± 24.3	61.3 ± 23.9	57.3 ± 26.4
*p*-value	0.434	0.122	0.487	0.090	0.023	0.252
Metastasis, mean ± SD						
Yes	62.1 ± 23.2	50 ± 21.6	53.8 ± 28.6	56.4 ± 27.3	61.4 ± 24.3	51.5 ± 25.1
No	65.4 ± 18.2	73.7 ± 16.9	71.7 ± 22.4	71.7 ± 19.3	76.3 ± 21.6	68.3 ± 21.6
*p*-value	0.882	<0.001	0.016	0.076	0.029	0.022

^a^ *p* value based on the Kruskal–Wallis or Mann–Whitney test. ^b^ For the functional scales, higher scores indicate better functioning. USD 1 = AED 3.67.

**Table 4 ijerph-22-00671-t004:** Symptom scales of the QLQ-C30 by independent variables (N = 72) ^a^.

**Characteristic**	**Fatigue ^b^**	**Nausea and Vomiting ^b^**	**Pain ^b^**	**Dyspnea ^b^**	**Insomnia ^b^**	**Appetite Loss ^b^**	**Constipation ^b^**	**Diarrhea ^b^**	**Financial Difficulties ^b^**
Age (years), mean ± SD									
24–34	47.4 ± 22.8	31.1 ± 28.1	42.2 ± 21.7	31.1 ± 32	42.2 ± 29.5	31.1 ± 26.6	35.6 ± 32	26.7 ± 33.8	28.9 ± 27.8
35–44	32.6 ± 25.2	15.6 ± 20.2	33.3 ± 24	22.6 ± 31.5	33.3 ± 28.5	16.1 ± 24.1	28 ± 29.9	17.2 ± 29.7	35.5 ± 35.4
45–54	42.6 ± 29.1	31.9 ± 24.1	38.9 ± 32.8	27.8 ± 23.9	30.6 ± 17.2	47.2 ± 38.8	38.9 ± 37.2	13.9 ± 17.2	50 ± 36.2
54–72	54 ± 31.7	41.7 ± 32.5	50 ± 34.6	23.8 ± 24.2	64.3 ± 38	47.6 ± 36.3	52.4 ± 42.8	35.7 ± 38	45.2 ± 38.4
*p* value	0.054	0.021	0.299	0.686	0.027	0.005	0.303	0.231	0.387
Time since diagnosis (years), mean ± SD									
<1	43.9 ± 24.1	28.9 ± 23.5	41.2 ± 21.1	45.6 ± 29.8	43.9 ± 27.3	31.6 ± 23.5	36.8 ± 29.2	24.6 ± 29.1	45.6 ± 29.8
1–2	41.9 ± 27.1	26.2 ± 27.2	36.7 ± 27.9	16.2 ± 22	41.9 ± 29.5	29.5 ± 35	37.1 ± 38.6	21 ± 33.4	31.4 ± 35.2
2–3	37 ± 33.3	13 ± 20	40.7 ± 34.5	11.1 ± 16.7	29.6 ± 42.3	18.5 ± 33.8	18.5 ± 24.2	14.8 ± 33.8	48.1 ± 41.2
3–4	25.9 ± 35.7	33.3 ± 33.3	50 ± 44.1	55.6 ± 50.9	44.4 ± 38.5	33.3 ± 33.3	33.3 ± 33.3	44.4 ± 38.5	33.3 ± 33.3
4–5	44.4 ± 50.9	44.4 ± 50.9	44.4 ± 50.9	11.1 ± 19.2	44.4 ± 50.9	44.4 ± 50.9	44.4 ± 50.9	22.2 ± 19.2	33.3 ± 57.7
>5	48.1 ± 17	33.3 ± 33.3	38.9 ± 9.6	33.3 ± 33.3	33.3 ± 33.3	55.6 ± 38.5	66.7 ± 33.3	22.2 ± 19.2	55.6 ± 19.2
*p* value	0.738	0.602	0.887	0.005	0.774	0.466	0.443	0.576	0.358
Marital status, mean ± SD									
Single	48.9 ± 23	20 ± 24.6	45 ± 26.1	13.3 ± 23.3	36.7 ± 29.2	33.3 ± 31.4	43.3 ± 31.6	20 ± 28.1	43.3 ± 35.3
Married	39.2 ± 26.9	25.2 ± 23.3	37.9 ± 26.3	29.7 ± 29.9	39.4 ± 30.8	29.1 ± 31.5	33.3 ± 33.3	21.2 ± 29.7	38.2 ± 33.6
Divorced	49.2 ± 38.4	47.6 ± 45.6	42.9 ± 40.7	9.5 ± 16.3	57.1 ± 37.1	38.1 ± 44.8	47.6 ± 50.4	33.3 ± 47.1	33.3 ± 47.1
*p* value	0.407	0.375	0.588	0.068	0.392	0.852	0.540	0.917	0.661
Education level, mean ± SD									
Illiterate	70.4 ± 28	33.3 ± 33.3	50 ± 50	33.3 ± 33.3	22.2 ± 19.2	44.4 ± 50.9	66.7 ± 33.3	11.1 ± 19.2	55.6 ± 50.9
Primary and preparatory school	57.4 ± 35.7	50 ± 29.3	52.8 ± 32.4	25 ± 25.1	63.9 ± 41.3	66.7 ± 34.8	55.6 ± 47.8	36.1 ± 41.3	50 ± 36.2
Secondary school	39.5 ± 22	18.5 ± 21.3	34.3 ± 22.5	22.2 ± 25.6	40.7 ± 24.4	25.9 ± 29.3	25.9 ± 24.4	20.4 ± 28.3	38.9 ± 30.8
University undergraduate	38.7 ± 23.7	20.7 ± 24.3	37.9 ± 23.5	23 ± 29.7	32.2 ± 27.4	20.7 ± 22.6	32.2 ± 31.5	17.2 ± 29	29.9 ± 33.7
University graduate	25.6 ± 25.7	28.3 ± 26.1	33.3 ± 33.3	36.7 ± 36.7	43.3 ± 31.6	20 ± 28.1	33.3 ± 35.1	26.7 ± 30.6	43.3 ± 38.7
*p* value	0.050	0.025	0.336	0.750	0.117	0.004	0.227	0.540	0.374
Employment, mean ± SD									
Yes	38.5 ± 25.7	19.5 ± 24.7	36.2 ± 24.7	21.1 ± 28.6	34.1 ± 28.4	19.5 ± 25.8	30.9 ± 32	16.3 ± 29.9	30.9 ± 34.5
No	45.5 ± 29.7	36 ± 26.9	43.5 ± 30.9	31.2 ± 28.5	49.5 ± 33.2	45.2 ± 35	43 ± 37.7	30.1 ± 31.5	48.4 ± 33.2
*p* value	0.351	0.007	0.277	0.098	0.044	0.001	0.182	0.015	0.018
Monthly income (AED), mean ± SD									
<10,000	46 ± 27.2	32.2 ± 25.5	43.9 ± 28.1	30.3 ± 29.5	46.2 ± 32.3	37.9 ± 32.6	41.7 ± 36	31.1 ± 33.3	50.8 ± 35.6
10,000–20,000	37.8 ± 27.3	15 ± 25.4	40 ± 26.3	30 ± 33.1	36.7 ± 18.9	10 ± 16.1	26.7 ± 26.3	6.7 ± 14.1	20 ± 23.3
20,000–30,000	42 ± 21.4	25.9 ± 22.2	33.3 ± 16.7	18.5 ± 24.2	40.7 ± 27.8	33.3 ± 33.3	37 ± 35.1	7.4 ± 14.7	25.9 ± 27.8
>30,000	23.5 ± 31.2	13 ± 33.1	22.2 ± 31.2	3.7 ± 11.1	18.5 ± 33.8	14.8 ± 33.8	18.5 ± 33.8	11.1 ± 33.3	11.1 ± 16.7
*p* value	0.064	0.028	0.053	0.046	0.069	0.015	0.206	0.008	0.001
Physical activity, mean ± SD									
Yes	33.7 ± 25.8	20 ± 25.5	30 ± 22.1	21 ± 26.9	31.4 ± 26.7	21.9 ± 29.1	27.6 ± 30.8	11.4 ± 24.2	29.5 ± 31.1
No	48.9 ± 27.4	32.9 ± 26.8	48.2 ± 29.6	29.7 ± 30.2	49.5 ± 33	38.7 ± 33.8	44.1 ± 36.9	32.4 ± 33.8	46.8 ± 36.4
*p* value	0.018	0.026	0.005	0.209	0.015	0.022	0.053	0.001	0.036
Smoking habits, mean ± SD									
Smoker	41.9 ± 27.3	34.6 ± 22	34.6 ± 15.9	38.5 ± 32.9	43.6 ± 31.6	35.9 ± 34.6	33.3 ± 33.3	20.5 ± 29	43.6 ± 31.6
Nonsmoker	41.6 ± 28.7	24.8 ± 28.3	40.6 ± 29.5	21.2 ± 27.5	40 ± 32.3	29.1 ± 33.4	37.6 ± 36.3	22.4 ± 32.7	38.2 ± 36.5
Previous smoker	38.9 ± 11.1	25 ± 16.7	37.5 ± 34.4	41.7 ± 16.7	41.7 ± 16.7	33.3 ± 0	25 ± 16.7	25 ± 16.7	25 ± 16.7
*p* value	0.985	0.297	0.899	0.047	0.929	0.591	0.869	0.715	0.629
Children, mean ± SD									
Yes	38.5 ± 27.5	27 ± 27	38.8 ± 29.4	27 ± 28.9	37.9 ± 30.2	29.3 ± 31.9	36.2 ± 36	19.5 ± 28.6	36.8 ± 34.6
No	54 ± 25	25 ± 26.8	41.7 ± 19.3	19 ± 28.4	52.4 ± 33.9	35.7 ± 35.7	35.7 ± 30.6	33.3 ± 39.2	45.2 ± 36.1
*p* value	0.043	0.894	0.321	0.281	0.097	0.550	0.870	0.263	0.393
Stage, mean ± SD									
0	42 ± 27.1	31.5 ± 15.5	42.6 ± 23.7	25.9 ± 22.2	33.3 ± 23.6	37 ± 30.9	37 ± 30.9	18.5 ± 24.2	44.4 ± 28.9
I	37.4 ± 24.6	21.2 ± 25.3	32.6 ± 14.1	16.7 ± 22.4	33.3 ± 30.9	33.3 ± 38.5	33.3 ± 34.1	15.2 ± 26.7	31.8 ± 30
II	42.2 ± 30.3	22.2 ± 33.1	35.6 ± 27.4	15.6 ± 27.8	44.4 ± 37.1	22.2 ± 30	44.4 ± 41.1	20 ± 37.4	51.1 ± 39.6
III	41.9 ± 28	41 ± 27.7	51.3 ± 35	41 ± 30.9	51.3 ± 25.9	30.8 ± 31.8	35.9 ± 28.7	38.5 ± 35.6	23.1 ± 31.6
IV	62.2 ± 12.7	26.7 ± 25.3	53.3 ± 39.8	46.7 ± 29.8	53.3 ± 38	53.3 ± 18.3	40 ± 43.5	20 ± 18.3	66.7 ± 33.3
*p* value	0.310	0.172	0.396	0.021	0.481	0.270	0.995	0.196	0.052
Radiotherapy, mean ± SD									
Yes	43.6 ± 35	21.3 ± 30.6	46 ± 35.8	12 ± 19	44 ± 39.3	26.7 ± 38.5	34.7 ± 39.1	22.7 ± 36.9	37.3 ± 43.4
No	40.4 ± 23	29.4 ± 24.4	35.8 ± 21.7	32.6 ± 30.7	39 ± 26.3	32.6 ± 29.1	36.9 ± 32.8	22 ± 28	39 ± 29.7
*p* value	0.811	0.088	0.438	0.004	0.842	0.162	0.585	0.586	0.396
Chemotherapy, mean ± SD									
Yes	37.6 ± 28	25.2 ± 22.2	37.2 ± 28.3	28.4 ± 29.5	37.6 ± 29.2	27.7 ± 29.7	33.3 ± 31.1	18.4 ± 26.7	39.7 ± 34.5
No	48.9 ± 25.7	29.3 ± 34.1	43.3 ± 26.4	20 ± 27.2	46.7 ± 34.7	36 ± 37.2	41.3 ± 41.1	29.3 ± 37.7	36 ± 35.9
*p* value	0.096	0.966	0.272	0.212	0.241	0.439	0.568	0.316	0.600
Metastasis, mean ± SD									
Yes	51.5 ± 31.9	38.6 ± 29.3	48.5 ± 34.5	28.8 ± 25.8	50 ± 32.1	47 ± 33.6	45.5 ± 33.4	19.7 ± 26.5	45.5 ± 35
No	35.3 ± 23.7	17.5 ± 21.3	34.2 ± 22.3	20.8 ± 28.9	35.8 ± 31.5	20.8 ± 30.8	30 ± 35.2	20 ± 32.7	37.5 ± 36.3
*p* value	0.037	0.004	0.191	0.146	0.119	0.002	0.057	0.623	0.325

^a^ *p* values were determined via the Kruskal–Wallis test or the Mann–Whitney test. ^b^ For the symptom scales, higher scores indicate worse functioning. USD 1 = AED 3.67.

**Table 5 ijerph-22-00671-t005:** Symptom scales of the QLQ-CX24 by independent variables (N = 72) ^a^.

Characteristics	Symptom Experience ^b^	Body Image ^b^	Sexual/Vaginal Functioning (*n* = 44) ^b^	Lymphedema ^b^	Peripheral Neuropathy ^b^	Menopausal Symptoms ^b^	Sexual Worry ^b^
Age (years), mean ± SD							
24–34	36.6 ± 17.8	39.3 ± 21.4	28.3 ± 21.6	22.2 ± 30	24.4 ± 29.5	26.7 ± 28.7	28.9 ± 30.5
35–44	24.9 ± 16.7	39.8 ± 31.0	41.2 ± 31.4	12.9 ± 18.6	23.7 ± 26.1	35.5 ± 24.2	43 ± 31.3
45–54	32.8 ± 16.3	42.6 ± 35.7	29.8 ± 20.4	52.8 ± 33.2	50 ± 26.6	47.2 ± 33.2	36.1 ± 38.8
54–72	29.7 ± 17.6	34.9 ± 38.9	3.10 ± 8.8	33.3 ± 43.4	31 ± 38	23.8 ± 27.5	7.1 ± 19.3
*p* value	0.176	0.837	0.005	0.007	0.049	0.124	0.003
Time since diagnosis (years), mean ± SD							
<1	35.3 ± 15.4	34.5 ± 25.4	30.9 ± 27.4	31.6 ± 30.4	26.3 ± 28.5	40.4 ± 30.6	38.6 ± 33.8
1–2	27.9 ± 15.3	38.1 ± 31.6	27.7 ± 28.4	21.9 ± 31.3	32.4 ± 30.8	31.4 ± 24.2	26.7 ± 34.1
2–3	24.9 ± 21.9	51.9 ± 41.2	37.5 ± 34.3	25.9 ± 40.1	37 ± 35.1	37 ± 38.9	37 ± 30.9
3–4	38.4 ± 34.5	40.7 ± 28.0	25.0 ± –	33.3 ± 33.3	33.3 ± 33.3	22.2 ± 38.5	55.6 ± 19.2
4–5	24.2 ± 18.4	55.6 ± 38.5		11.1 ± 19.2	0 ± 0	33.3 ± 0	11.1 ± 19.2
>5	24.2 ± 21.2	25.9 ± 35.7		33.3 ± 57.7	22.2 ± 38.5	11.1 ± 19.2	38.6 ± 33.8
*p* value	0.550	0.803	0.898	0.752	0.425	0.541	0.291
Marital status, mean ± SD							
Single	38.5 ± 19.7	53.3 ± 31.8	38.9 ± 38.6	29.2 ± 9.2	33.3 ± 41.6	30 ± 33.1	23.3 ± 38.7
Married	28.7 ± 16.2	36.6 ± 31.0	29.4 ± 27.5	33.8 ± 4.6	30.9 ± 29.3	34.5 ± 28.7	36.4 ± 32.2
Divorced	23.8 ± 20.7	39.7 ± 31.3	16.6 ± 20.4	26.2 ± 9.9	14.3 ± 17.8	28.6 ± 12.6	9.5 ± 16.3
*p* value	0.287	0.279	0.563	0.511	0.394	0.789	0.046
Education level, mean ± SD							
Illiterate	35.4 ± 17.8	40.7 ± 52.5	41.6 ± –	55.6 ± 50.9	44.4 ± 38.5	44.4 ± 50.9	33.3 ± 57.7
Primary and preparatory school	37.1 ± 15.3	42.6 ± 40.2	20.8 ± 29.9	30.6 ± 46	41.7 ± 32.2	44.4 ± 29.6	11.1 ± 29.6
Secondary school	28.1 ± 18.9	30.9 ± 22.4	30.8 ± 24.5	35.2 ± 31.3	25.9 ± 33.4	31.5 ± 31.3	27.8 ± 34.8
University undergraduate	27.7 ± 14.7	44.1 ± 29.8	38.8 ± 29.0	14.9 ± 22.9	25.3 ± 26.2	32.2 ± 22.7	41.4 ± 29.1
University graduate	27.0 ± 23.5	35.6 ± 34.3	8.3 ± 11.8	23.3 ± 27.4	30 ± 33.1	23.3 ± 27.4	36.7 ± 29.2
*p* value	0.313	0.789	0.068	0.162	0.447	0.425	0.023
Employment, mean ± SD							
Yes	26.5 ± 17.4	40.9 ± 30.4	37.3 ± 32.1	17.9 ± 24.8	28.5 ± 30.3	29.3 ± 20	38.2 ± 33
No	33.6 ± 16.7	36.9 ± 32.8	22.4 ± 21.3	35.5 ± 38.4	31.2 ± 31	38.7 ± 35.6	23.7 ± 31.3
*p* value	0.046	0.537	0.130	0.056	0.684	0.420	0.042
Monthly income (AED), mean ± SD							
<10,000	34.2 ± 17.5	45.5 ± 33.4	28.6 ± 27.5	34.1 ± 33.3	32.1 ± 4.8	34.8 ± 31.3	29.5 ± 34.6
10,000–20,000	23.9 ± 13.2	31.1 ± 22.1	58.3 ± 26.3	10 ± 22.5	30.6 ± 9.7	40 ± 21.1	50 ± 28.3
20,000–30,000	28.6 ± 11.6	38.3 ± 28.9	22.9 ± 15.7	18.5 ± 33.8	24.2 ± 8.1	33.3 ± 23.6	29.6 ± 35.1
>30,000	14.5 ± 16.9	18.5 ± 22.9	0 ± 0	7.4 ± 22.2	24.2 ± 8.1	18.5 ± 17.6	25.9 ± 22.2
*p* value	0.011	0.087	0.071	0.015	0.280	0.283	0.212
Physical activity, mean ± SD							
Yes	22.6 ± 14.8	35.2 ± 26.6	28.9 ± 29.5	14.3 ± 21.8	18.1 ± 21.9	25.7 ± 19.9	32.4 ± 32.8
No	36.2 ± 17.1	42.9 ± 35.1	29.9 ± 27.1	36 ± 37.2	40.5 ± 33.5	40.5 ± 32.5	31.5 ± 33.3
*p* value	< 0.001	0.417	0.769	0.011	0.003	0.056	0.881
Smoking habits, mean ± SD							
Smoker	34.7 ± 14.1	47.9 ± 28.1	36.1 ± 24.7	12.8 ± 21.7	25.6 ± 24.2	33.3 ± 19.2	56.4 ± 34.4
Nonsmoker	28.3 ± 18.1	38.4 ± 32.6	28.0 ± 29.2	29.1 ± 34.6	31.5 ± 32.3	32.7 ± 29	26.1 ± 31.2
Previous smoker	31.1 ± 16.3	22.2 ± 12.8	25.0 ± 23.6	16.7 ± 19.2	16.7 ± 19.2	41.7 ± 41.9	33.3 ± 0
*p* value	0.501	0.328	0.521	0.310	0.686	0.873	0.012
Children, mean ± SD							
Yes	27.8 ± 15.9	37.5 ± 31.5	27.4 ± 25.3	26.4 ± 33.5	29.9 ± 30.4	33.3 ± 26.5	33.9 ± 32.7
No	37.0 ± 21.3	46.0 ± 30.5	38.0 ± 36.4	21.4 ± 28.1	28.6 ± 31.6	33.3 ± 34.6	23.8 ± 33.1
*p* value	0.121	0.288	0.565	0.707	0.855	0.733	0.237
Stage, mean ± SD							
0	39.7 ± 17.8	30.9 ± 31.8	22.9 ± 17.1	40.7 ± 36.4	40.7 ± 32.4	48.1 ± 37.7	22.2 ± 33.3
I	26.9 ± 16.7	43.4 ± 28.9	44.4 ± 28.9	18.2 ± 33.7	25.8 ± 25.1	31.8 ± 24.1	37.9 ± 33
II	23.8 ± 14.6	45.9 ± 34.9	53.3 ± 36.6	13.3 ± 21.1	11.1 ± 16.3	28.9 ± 27.8	33.3 ± 35.6
III	35.4 ± 15.8	35.0 ± 22.2	25 ± 26.4	28.2 ± 32.9	41 ± 30.9	38.5 ± 18.5	38.5 ± 35.6
IV	37.0 ± 23.2	51.1 ± 47.5	21.7 ± 26.7	40 ± 27.9	40 ± 54.8	26.7 ± 43.5	33.3 ± 23.6
*p* value	0.106	0.724	0.245	0.104	0.056	0.354	0.708
Radiotherapy, mean ± SD							
Yes	25.1 ± 19.0	42.7 ± 38.0	26.8 ± 34.3	17.3 ± 33.5	32 ± 35.3	30.7 ± 30.3	29.3 ± 36.4
No	32.0 ± 16.1	37.4 ± 27.3	30.8 ± 24.7	29.8 ± 31.3	28.4 ± 27.8	34.8 ± 26.9	33.3 ± 31.1
*p* value	0.111	0.796	0.266	0.037	0.885	0.427	0.416
Chemotherapy, mean ± SD							
Yes	28.2 ± 17.0	40.7 ± 31.6	29.2 ± 27.9	27.7 ± 31.3	31.9 ± 31.8	28.4 ± 23	38.3 ± 33.3
No	32.2 ± 18.0	36.4 ± 31.0	30.1 ± 28.3	21.3 ± 34.5	25.3 ± 27.7	42.7 ± 34	20 ± 28.9
*p* value	0.235	0.513	0.884	0.234	0.438	0.086	0.017
Metastasis, mean ± SD							
Yes	33.9 ± 17.2	47.0 ± 35.0	12.5 ± 23.4	33.3 ± 37.1	36.4 ± 35.5	34.8 ± 26.2	33.3 ± 32.5
No	24.6 ± 15.5	36.1 ± 31.3	29.9 ± 30.2	16.7 ± 28.2	24.2 ± 26.1	30 ± 27	32.5 ± 35.8
*p* value	0.044	0.259	0.101	0.037	0.216	0.425	0.791

^a^ *p* value was determined via the Kruskal–Wallis test or the Mann–Whitney test. ^b^ For the symptom scales, higher scores indicate worse functioning. USD 1 = AED 3.67.

**Table 6 ijerph-22-00671-t006:** Functional items on the QLQ-CX24 by independent variables ^a^.

Characteristic	Sexual Activity ^b^ (*n* = 72)	Sexual Enjoyment ^b^ (*n* = 47)
Age (years), mean ± SD		
24–34	33.3 ± 33.3	26.7 ± 30.6
35–44	33.3 ± 24.3	28.3 ± 29.2
45–54	22.2 ± 25.9	40.7 ± 27.8
54–72	9.5 ± 20.4	12.5 ± 35.4
*p* value	0.023	0.143
Time since diagnosis (years), mean ± SD		
<1	29.8 ± 29.2	33.3 ± 32.2
1–2	21.9 ± 25.5	25.6 ± 31.7
2–3	22.2 ± 23.6	16.7 ± 19.2
3–4	66.7 ± 0	33.3 ± 32.2
4–5	33.3 ± 33.3	–
>5	33.3 ± 33.3	–
*p* value	0.169	0.739
Marital status, mean ± SD		
Single	23.3 ± 38.7	27.8 ± 32.8
Married	29.1 ± 24.9	25.9 ± 28.9
Divorced	14.3 ± 26.2	40 ± 43.5
*p* value	0.208	0.776
Education level, mean ± SD		
Illiterate	0 ± 0	0 ± 0
Primary and preparatory school	8.3 ± 15.1	8.3 ± 15.1
Secondary school	25.9 ± 29.3	25.9 ± 29.3
University undergraduate	34.5 ± 25.9	34.5 ± 25.9
University graduate	36.7 ± 29.2	36.7 ± 29.2
*p* value	0.010	0.496
Employment, mean ± SD		
Yes	33.3 ± 25.8	21.2 ± 24.2
No	18.3 ± 27	33.3 ± 34.7
*p* value	0.010	0.272
Monthly income (AED), mean ± SD		
<10,000	21.2 ± 27	28.7 ± 32
10,000–20,000	43.3 ± 22.5	41.7 ± 31.9
20,000–30,000	29.6 ± 26.1	20 ± 18.3
>30,000	33.3 ± 28.9	0 ± 0
*p* value	0.066	0.384
Physical activity, mean ± SD		
Yes	28.6 ± 25.7	22.8 ± 29.5
No	25.2 ± 28.8	31 ± 31.3
*p* value	0.480	0.358
Smoking habits, mean ± SD		
Smoker	41 ± 14.6	23.3 ± 22.5
Nonsmoker	23.6 ± 29.2	30.4 ± 33.2
Previous smoker	25 ± 16.7	11.1 ± 19.2
*p* value	0.044	0.604
Children, mean ± SD		
Yes	28.2 ± 26.3	28.1 ± 30.5
No	21.4 ± 31	25.9 ± 32.4
*p* value	0.279	0.816
Stage, mean ± SD		
0	22.2 ± 33.3	37.5 ± 27.8
I	33.3 ± 23	24.2 ± 21.6
II	22.2 ± 27.2	33.3 ± 42.2
III	33.3 ± 30.4	18.5 ± 29.4
IV	26.7 ± 27.9	33.3 ± 33.3
*p* value	0.519	0.659
Radiotherapy, mean ± SD		
Yes	17.3 ± 23.8	14.3 ± 31.3
No	31.9 ± 27.8	33.3 ± 28.9
*p* value	0.029	0.019
Chemotherapy, mean ± SD		
Yes	30.5 ± 25.8	33.3 ± 30.9
No	20 ± 28.9	18.5 ± 28.5
*p* value	0.067	0.082
Metastasis, mean ± SD		
Yes	30.3 ± 27	29.2 ± 27.8
No	21.7 ± 24.5	23.3 ± 31.7
*p* value	0.215	0.456

^a^ *p* value was determined via the Kruskal–Wallis test or the Mann–Whitney test. ^b^ For the functional items, higher scores indicate better functioning. USD 1 = AED 3.67.

**Table 7 ijerph-22-00671-t007:** Relevant factors influencing the global health status of CC survivors according to multiple regression analysis (*n* = 72).

Variable	β	*p* Value
Intercept	90.876	<0.001
Marital status	−0.053	0.656
Education level	0.153	0.268
Employment	0.142	0.316
Income (AED)	0.502	<0.001
Cancer stage	0.158	0.180
Time since diagnosis (years)	−0.137	0.257
Radiotherapy	−0.133	0.272
Chemotherapy	−0.055	0.624

β: standardized coefficient beta; *p*: significance.

## Data Availability

Data are available upon reasonable request due to privacy.

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
