# Peer review of "Quality of Life of Emirati Women with Cervical Cancer Using EORTC QLQ-30 and CX24: A First Look in the UAE"

_ijerph, 2025, doi:10.3390/ijerph22050671_

Round 1

Reviewer 1 Report

Comments and Suggestions for Authors

Thank you for the opportunity to review the manuscript entitled “Quality of Life of Emirati Women with Cervical Cancer Using EORTC QLQ-30 and CX24: A First Look in the UAE.” This article's intent is appreciated, as many women around the world experience cervical cancer. This is an important article in that it displays the cultural differences found between QOL factors in Emirati cervical cancer patients, and how these differences should be taken into consideration when developing initiatives targeted towards improving QOL in these patients. Overall, I found this article to be informative and interesting as cervical cancer remains a high burden for many women around the world. Please see the below comments for the manuscript.  

  • The abstract is clear and concise. Was there a reason that ANOVA and independent sample t-test weren’t listed in the abstract? Based on the methods in 3.4, these were the primary statistical methods with the Kruskal–Wallis test and Mann–Whitney test being secondary. 
  • The introduction contains a well-rounded overview of the current literature surrounding this topic. I am unsure if the separate section for “Literature Review” is necessary. I think subsections within the introduction, similar to the methodology, would potentially be a better way to present the information in a manner that better orients the reader to the purpose of this study. 
  • In the methods section:
    • The ethics approval and consent to participate section feels unneccessary since the back matter contains statements that contain the majority of this information. 
    • Section 3.4 presents unclearly, particularly around the scale scoring and the meaning associated with it. There is a lot of repetition in the words, but it is unclear to me how this was transformed and what factors are using the scaled format. I think clarity in this area would strengthen this article. 
  • A lot of the written version of the results is presenting the same information as the tables at this time, particularly in 4.1 and 4.2. Scaling this back or utilizing the results section to better elaborate on items of importance would make this manuscript stronger. 
  • One issue seen periodically throughout the entire manuscript is the use of first-person language instead of third-person language. Additionally, this manuscript feels very long and dense, and I think some streamlining of information would be of great benefit. 
  • Table 1: Tablets under treatment method should fall under chemotherapy. 
  • Table 2: The switching from functional scales to symptom scales and then back to functional scales could present readers with confusion, especially since the scores have different meaning. I think creating two separate tables or clearly delineating where the scale changes would make this more clear to the reader. 
  • Table 7: According to the key, the “beta” being displayed is a standardized coefficient beta, but when looking at the data in the table, it appears that these are unstandardized coefficient betas. A second look at this analysis and the associated results is needed.

Author Response

Comments and Suggestions for Authors

Thank you for the opportunity to review the manuscript entitled “Quality of Life of Emirati Women with Cervical Cancer Using EORTC QLQ-30 and CX24: A First Look in the UAE.” This article's intent is appreciated, as many women around the world experience cervical cancer. This is an important article in that it displays the cultural differences found between QOL factors in Emirati cervical cancer patients, and how these differences should be taken into consideration when developing initiatives targeted towards improving QOL in these patients. Overall, I found this article to be informative and interesting as cervical cancer remains a high burden for many women around the world. Please see the below comments for the manuscript.  

Response:

Thank you for your thoughtful review and for recognizing the importance and relevance of our study, particularly regarding the cultural factors affecting the Quality of Life in Emirati women with cervical cancer. Your comments are VERY valuable and have significantly helped us improve the clarity and impact of our manuscript. We have carefully addressed each of your detailed suggestions in the responses below.

  • The abstract is clear and concise. Was there a reason that ANOVA and independent sample t-test weren’t listed in the abstract? Based on the methods in 3.4, these were the primary statistical methods with the Kruskal–Wallis test and Mann–Whitney test being secondary. 

Response:

We appreciate this valuable remark, Indeed, ANOVA and independent-sample t-tests were our primary methods. We have revised the abstract to explicitly state all statistical methods clearly. 

  • The introduction contains a well-rounded overview of the current literature surrounding this topic. I am unsure if the separate section for “Literature Review” is necessary. I think subsections within the introduction, similar to the methodology, would potentially be a better way to present the information in a manner that better orients the reader to the purpose of this study. 

Response:

Thank you for this insightful comment. We have restructured the paper by integrating the Literature Review section into the Introduction. Clear subsections have been created to improve readability and orientation for the reader. This section now provides a logical flow that contextualizes the importance of our study clearly.

  • In the methods section: The ethics approval and consent to participate section feels unneccessary since the back matter contains statements that contain the majority of this information. 

Response:

Thank you for highlighting this redundancy. We have removed the detailed ethics section from the methods and replaced it with a brief reference to the approval within the methods text.

  • In the methods section: Section 3.4 presents unclearly, particularly around the scale scoring and the meaning associated with it. There is a lot of repetition in the words, but it is unclear to me how this was transformed and what factors are using the scaled format. I think clarity in this area would strengthen this article. 

Response:

Thank you for highlighting the lack of clarity and repetition in section 3.4 (new 2.4). We have clarified the explanation of scale scoring and interpretation in alignment with the official EORTC guidelines. Specifically, we detailed the process of linear transformation and clearly stated how the scores were interpreted, which significantly improves readability and precision.

  • A lot of the written version of the results is presenting the same information as the tables at this time, particularly in 4.1 and 4.2. Scaling this back or utilizing the results section to better elaborate on items of importance would make this manuscript stronger. 

Response:

Thank you for your recommendation. We revised section 3.1 and 3.1 (old 4.1 and 4.2) by reducing repetition from the tables and focusing the narrative on key highlights and clinically relevant demographic and clinical characteristics. We applied the same approach to sections 3.3 and 3.4, emphasizing only essential numerical results and statistically significant findings. This provides a clearer and more concise presentation of the participant characteristics, strengthening the manuscript’s readability as suggested.

  • One issue seen periodically throughout the entire manuscript is the use of first-person language instead of third-person language. Additionally, this manuscript feels very long and dense, and I think some streamlining of information would be of great benefit. 

Response:

Thank you for highlighting these points. Regarding the use of first-person language, we carefully reviewed the manuscript and confirmed that it already consistently adheres to the third-person academic writing convention; one first-person ("we") usage was found in the Acknowledgments, this was changed.

Regarding the length and density of the manuscript, sections 3.1 to 3.4 have now been significantly streamlined, reducing redundancy and focusing clearly on the most clinically relevant and statistically significant findings. 

  • Table 1: Tablets under treatment method should fall under chemotherapy. 

Response:

Thank you for your insightful comment. We have updated Table 1 by merging the "Tablets" category clearly under Chemotherapy to accurately represent oral chemotherapy treatments. We adjusted the numbers and percentages accordingly.

  • Table 2: The switching from functional scales to symptom scales and then back to functional scales could present readers with confusion, especially since the scores have different meaning. I think creating two separate tables or clearly delineating where the scale changes would make this more clear to the reader. 

Response:

Thank you for highlighting this issue. Table 2 has now been clearly delineated into two separate sections: one for the QLQ-C30 scale and another for the QLQ-CX24 scale. Each scale includes its own functional and symptom items. This separation should clarify the differences in score interpretations and enhance readability for the readers. Unfortunately, we cannot merge the functional and symptom scales from the two instruments, as they contain entirely different items. We have maintained the same order as presented in the official EORTC scales.

  • Table 7: According to the key, the “beta” being displayed is a standardized coefficient beta, but when looking at the data in the table, it appears that these are unstandardized coefficient betas. A second look at this analysis and the associated results is needed.

Response:

Thank you so much for highlighting this discrepancy. Upon re-examination, we confirmed that the coefficients displayed in Table 7 were indeed unstandardized betas rather than standardized betas. We have corrected the table with the standardized Coefficient Betas and adjusted the manuscript accordingly to maintain analytical clarity and accuracy

Reviewer 2 Report

Comments and Suggestions for Authors

Dear Authors,

In this referenced paper titled “Quality of Life of Emirati Women with Cervical Cancer Using EORTC QLQ-30 and CX24: A First Look in the UAE” the study demonstrated factors influencing the QoL of Emirati women with CC. The obtained results can influence the creation of a strategy for a better understanding of key points of QoLcervical cancer patients.

Overall, the article is well written and organized. This is an interesting paper containing useful results. However, to strengthen the quality of the current manuscript, there are several points that need to be further addressed, and the quality needs improvement as described below before considering accepting this manuscript for publication.

  • Put a more recent reference for the incidence of cervical cancer.
  • Systematize the data in the introductory part of the paper because many parts are repeated.
  • State the inclusion and exclusion criteria.
  • How many participants did not meet the criteria for participation in the research.
  • There is no need to repeat the results presented in the tables.
  • The article is generally too long, although the results are very interesting. Perhaps the advice to the authors is to consider presenting the results in two articles or to shorten the manuscript.
  • Edit references uniformly according to the magazine's propositions.

Author Response

Comments and Suggestions for Authors

In this referenced paper titled “Quality of Life of Emirati Women with Cervical Cancer Using EORTC QLQ-30 and CX24: A First Look in the UAE” the study demonstrated factors influencing the QoL of Emirati women with CC. The obtained results can influence the creation of a strategy for a better understanding of key points of QoLcervical cancer patients.

Overall, the article is well written and organized. This is an interesting paper containing useful results. However, to strengthen the quality of the current manuscript, there are several points that need to be further addressed, and the quality needs improvement as described below before considering accepting this manuscript for publication.

Response:

Thank you for your valuable comments and constructive feedback on our manuscript. We appreciate your recognition of the manuscript's strengths and its potential to contribute to better strategies for addressing the quality of life in Emirati women with cervical cancer. We have carefully reviewed and addressed each of your comments point by point, making the necessary revisions and clarifications as suggested. These adjustments significantly enhance the clarity, readability, and overall quality of the manuscript.

  • Put a more recent reference for the incidence of cervical cancer.

Response:

Thank you for this valuable comment. We have updated the manuscript with recent statistics on the incidence of cervical cancer, both globally and specifically for the UAE. The introduction section now includes the most recent references from the WHO (2022) and UAE National Cancer Registry (2021) to provide accurate and up-to-date context regarding cervical cancer burden.

  • Systematize the data in the introductory part of the paper because many parts are repeated.

Response:

Thank you for this valuable observation. The introduction has now been streamlined to remove repetitive content and improve flow. We reorganized and merged sections that previously overlapped, ensuring that each part clearly addresses different aspects of cervical cancer incidence, the significance of quality of life assessments, and the context within the UAE.

  • State the inclusion and exclusion criteria.

Response:

We have now explicitly stated these criteria in the methodology section under “2.1. Study Design and Sample.” The study included Emirati women aged ≥18 years diagnosed specifically with cervical cancer, while those diagnosed with other types of cancers were excluded.

  • How many participants did not meet the criteria for participation in the research.

Response:

Two initially recruited participants were excluded due to an additional cancer diagnosis, and this detail has also been included in the manuscript for clarity.

  • There is no need to repeat the results presented in the tables.
  • The article is generally too long, although the results are very interesting. Perhaps the advice to the authors is to consider presenting the results in two articles or to shorten the manuscript.

Response:

Thank you for pointing out these concerns. We have thoroughly revised the results sections (3.1, 3.2, 3.3, and 3.4) to remove repetitive information already presented in the tables, emphasizing only the key and clinically important findings in the narrative. This significantly shortened the manuscript, making the text clearer and easier to follow.

Regarding your suggestion of splitting the manuscript into two separate articles: we appreciate this advice. However, we prefer to keep these findings within a single paper to provide comprehensive insights about quality-of-life factors in Emirati cervical cancer survivors. We believe the revised, streamlined version addresses the issue effectively.

  • Edit references uniformly according to the magazine's propositions.

Response:

We have replaced the two references about the incidence of cervical cancer with the most recent ones. The other references' formatting follows IJERPH formatting.

Round 2

Reviewer 2 Report

Comments and Suggestions for Authors

Dear Authors,

Thank you for taking into account all the suggestions. The manuscript now looks much more transparent. I wish you the best of luck in your future work.

Reviewer